# White Tea Reduces Dyslipidemia, Inflammation, and Oxidative Stress in the Aortic Arch in a Model of Atherosclerosis Induced by Atherogenic Diet in ApoE Knockout Mice

**DOI:** 10.3390/ph17121699

**Published:** 2024-12-17

**Authors:** Merve Huner Yigit, Mehtap Atak, Ertugrul Yigit, Zehra Topal Suzan, Mehmet Kivrak, Huseyin Avni Uydu

**Affiliations:** 1Department of Medical Biochemistry, Faculty of Medicine, Recep Tayyip Erdogan University, 53000 Rize, Turkey; merve.huner@erdogan.edu.tr; 2Department of Medical Biochemistry, Faculty of Medicine, Karadeniz Technical University, 61080 Trabzon, Turkey; ertugrulyigit@ktu.edu.tr; 3Department of Histology and Embryology, Faculty of Medicine, Recep Tayyip Erdogan University, 53000 Rize, Turkey; zehra.suzan@erdogan.edu.tr; 4Department of Biostatistics, Faculty of Medicine, Recep Tayyip Erdogan University, 53000 Rize, Turkey; mehmet.kivrak@erdogan.edu.tr; 5Department of Medical Biochemistry, Faculty of Medicine, Samsun University, 55080 Samsun, Turkey; huseyin.uydu@samsun.edu.tr

**Keywords:** ApoE^−/−^ mice, aortic arch, atherosclerosis, inflammation, oxidative stress, white tea

## Abstract

**Objective:** In this study, we aimed to evaluate the potential effects of white tea (WT) in the atherosclerosis process characterized by oxidative stress, inflammation, and dyslipidemia. **Methods:** In our study, apolipoprotein E knockout (ApoE^−/−^) mice (RRID: IMSR_JAX:002052) and C57BL/6J mice (RRID: IMSR_JAX:000664) were used. In the atherosclerosis model induced by an atherogenic diet (AD), WT was administered via oral gavage at two different concentrations. The animals were sacrificed by decapitation under anesthesia, and their serum and aortic tissues were collected. Total cholesterol (TC), triglyceride (TG), interleukin (IL)-1β, IL-6, IL-10, IL-12, tumor necrosis factor-α (TNF-α), interferon-γ, myeloperoxidase, paraoxonase-1, lipoprotein-associated phospholipase A2, oxidized low-density lipoprotein (Ox-LDL), lectin-like oxidized LDL receptor (LOX-1), a disintegrin, and metalloprotease (ADAM) 10 and 17 activities were determined via colorimetric, enzyme-linked immunoassay, and fluorometric methods. **Results:** WT supplementation decreased serum Ox-LDL, LOX-1, TC, and TG levels by approximately 50%. TNF- and IL-6 levels were reduced by approximately 30% in the aortic arch. In addition, ADAM10/17 enzyme activities were found to be reduced by approximately 25%. However, no change in the AD-induced fibrotic cap structure was observed in the aortic root. **Conclusions:** The findings indicate that white tea effectively reduced oxidative stress, inflammation, and dyslipidemia in atherosclerosis but does not affect atheroma plaque morphology.

## 1. Introduction

Cardiovascular diseases (CVDs) stand as the leading cause of global mortality [1]. Atherosclerosis, a chronic inflammatory process in the arterial wall, is the primary culprit, exacerbated by dyslipidemia, diabetes mellitus, and hypertension [2]. It is marked by the formation of atheroma plaques in medium and large arteries. Oxidized low-density lipoprotein (Ox-LDL) molecules in the subendothelial region lead to foam cell formation, endothelial damage, and the prevalence of atherosclerotic lesions. Ox-LDL receptor-1 (OLR-1/LOX-1), tumor necrosis factor alpha (TNF-α), and atherogenic interleukins (IL-1β, IL-6, etc.) play an important role in the pathogenesis of atherosclerosis [3,4,5,6].

The main goal in preventing CVDs is to lower low-density lipoprotein cholesterol (LDL-C) with drugs usually referred to as statins. However, previous studies have emphasized the importance of inflammation in the pathogenesis of atherosclerosis. Although statin therapies exert a pleiotropic anti-inflammatory effect, inflammation is not the primary target, and residual cardiovascular risk remains high in statin-treated patients [7,8]. In addition, several adverse effects of various synthetic drugs used to treat cardiovascular diseases have been reported, including gastrointestinal reactions, hyperkalemia, and arhythmias [9,10]. Due to these negative effects of synthetic drugs, interest in natural products with wide structural diversity and biodiversity has increased [11]. Experimental research has shown that some antioxidant natural products and their active compounds can prevent and treat cardiovascular diseases through different mechanisms of action [12]. 

Tea, the most consumed beverage in the world with water, is derived from the young buds and leaves of the *Camellia sinensis* (L.) plant [13]. The tea plant is originally from southwest China but is also cultivated in various regions, particularly in the northeast of Turkey [14,15]. Tea can be categorized into different types, such as white (WT), green (GT), oolong, black (BT), and pu’erh teas. WT is less well-known in Western communities but is highly valued in Asia, and its flavor is even more accepted in Europe than GT [16]. White tea contains a variety of compounds, including gallic acid (GA), catechin (C), epicatechin (EC), gallocatechin (GC), epigallocatechin (EGC), epicatechin 3-gallate (ECG), and epigallocatechin 3-gallate (EGCG), as well as alkaloids (caffeine and theobromine), amino acids, proteins, chlorophyll, carbohydrates, volatile organic compounds, minerals, and trace elements [17]. This has sparked interest in its potential to promote healthy skin, a unique aspect that sets it apart from other teas. WT, which is produced with minimum processing, is distinguished from other types of tea by its bioactive components and contains high levels of ECGC. Additionally, white tea’s lipolytic activity and ability to inhibit adipogenesis have garnered particular attention, especially in developed countries grappling with significant increases in obesity and obesity-related diseases [18,19]. There is limited research in the literature on the potential effects of white tea on oxidative stress, dyslipidemia, and inflammation in the development of atherosclerosis.

The lack of effective treatments for coronary artery disease and stroke caused by atherosclerosis is a significant burden on society and governments. This study aimed to contribute to understanding the potential benefits of white tea for atherosclerosis and to investigate the potential of natural supplements as an alternative or complementary approach to existing treatments. To this end, the possible effects of white tea on inflammatory and oxidative stress biomarkers in atherosclerosis and atherosclerotic plaque burden were investigated. It is also envisioned that the findings may contribute to developing therapies to improve the lives of individuals affected by atherosclerosis. 

## 2. Results

### 2.1. HPLC Content and Total Polyphenol/Flavonoid Capacity of White Tea

The phenolic content of white tea was assessed via HPLC-DAD analysis, with the results presented in Table 1. In addition, the standard chormatograms are presented in Appendix A. While EGC and EGCG were recorded as the catechins detected at the highest rate in the white tea, C and EC catechins were not detected in the extract. Table 2 displays the findings of the total flavonoid and polyphenol capacity in the white tea administered in the research. 

### 2.2. Food Intake Results

The amount of feed consumed by the study groups for 16 weeks is presented in Table 3. According to this, there was no significant difference between the amount of feed consumed by ApoE^−/−^ mice fed with an atherogenic diet (*p* < 0.05).

### 2.3. Consuming White Tea Reduces the Rate of Weight Gain

To evaluate the effects of white tea given with the atherogenic diet on weight change, the animals were weighed every four weeks throughout the experiment. The weights of the groups are shown in Figure 1. There was no significant difference between the initial weights of the study groups (*p* > 0.05). However, the atherogenic diet (AD)-fed animals had significantly higher weights than the control diet (CD) diet-fed mice at weeks 4th, 8th, and 12th (*p* < 0.05). There was no significant difference between the AD-fed groups (*p* > 0.05). After four weeks of WT administration via oral gavage, at week 16, the weight of the animals in the WT100 and WT500 groups fed with AD was significantly lower than the case group (*p* < 0.05), while there was no statistically significant difference between the CD-fed sham group (*p* > 0.05).

### 2.4. White Tea Ameliorates AD-Induced Dyslipidemia

The routine biochemistry results measured in the serum obtained from the study groups are given in Table 4. As expected, AD-induced total cholesterol (TC) and triglyceride (TG) levels in ApoE^−/−^ mice were significantly higher than in CD diet-fed mice (*p* < 0.05). After WT administration, serum TC and TG levels were significantly decreased compared to the AD-fed case group (*p* < 0.05). There was no noteworthy difference between the groups in glucose (Glu), alanine transaminase (ALT), and aspartate transaminase (AST) measurements (*p* > 0.05). Blood urea nitrogen (BUN) value was significantly higher in all groups than in the control group (*p* < 0.05). 

### 2.5. White Tea Reduces Oxidative Stress Markers

As expected, Ox-LDL, LOX-1, and phospholipase-A2 (PLA2) levels, which play an important role in the onset and progression of atherosclerosis, increased significantly in the AD-fed case group compared to the CD-fed groups. In contrast, paraoxanase-1 (PON-1) levels decreased (*p* < 0.05). However, PON-1 levels increased after WT administration, while OxLDL, LOX-1, and Lp-PLA2 levels significantly decreased compared to the case group (*p* < 0.05). High-dose WT administration was significant only in PON-1 levels (*p* < 0.05) (Figure 2).

### 2.6. White Tea Reduces Inflammation Markers

In the AD-fed ApoE^−/−^ mice, the levels of atherogenic cytokines measured in both the serum and aortic arch were significantly higher than in the CD-fed mice (*p* < 0.05) (Figure 3). In addition to these cytokines, the activities of inflammation-related proteases, a disintegrin and metalloprotease (ADAM)10 and ADAM17 were also significantly increased by AD (*p* < 0.05) (Figure 4). After four weeks of WT administration to AD-fed ApoE^−/−^ mice, decreased TNF-α, interleukin (IL)-1β, IL-6, IL-12, and ADAM10 and ADAM17 enzyme activities were observed (*p* < 0.05). However, while there was no significant change in interferon-γ (IFN-γ) levels (*p* > 0.05), myeloperoxidase (MPO) levels were significantly lower in the WT500 group compared to the case group (*p* < 0.05). Anti-atherogenic cytokine IL-10 levels were significantly higher in the WT100 and WT500 groups than in the case group (*p* < 0.05).

### 2.7. WT Does Not Reduce Atherosclerotic Plaque Burden in the Aortic Root

In the control group, histopathologic evaluation revealed normal tunica intima, tunica media, and adventitia layers. No lesions were observed within the aortic root. Similarly, in the sham group, the morphologic structures of the tunica intima, tunica media, and tunica adventitia layers were also typical. However, in the examination of the vascular structures of the case group, an atherosclerotic lesion was detected in the tunica intima layer, along with a thin fibrotic cap structure. The cells in the lesion had a large cytoplasmic appearance, indicating a Type III atherosclerotic lesion. Additionally, the aortic root of the WT100 group showed the presence of an atherosclerotic lesion in the tunica intima, along with a thin fibrotic cap structure, indicating a Type III atherosclerotic lesion. Furthermore, in the aortic root of the WT500 group, atherosclerotic lesions were found in the intima layer, occasionally progressing into the tunica media, and were thought to be Type IV atherosclerotic lesions (Table 5) (Figure 5). 

## 3. Discussion

Atherosclerosis is characterized by chronic hyperlipidemia, inflammation, oxidative stress, and the gradual accumulation of LDL and fibrous molecules in focal areas of arteries [20]. It has been proven that oxidative stress, inflammation, and dyslipidemia play an essential role in the formation and occurrence of atherosclerotic plaque [21]. Hyperlipidemia is a vital risk factor for cardiovascular diseases, characterized by changes in the serum lipids profile, including high TG levels, high TC levels, and low high-density lipoprotein levels [22]. In our study, hyperlipidemic and hyperglycemic pictures emerged in the groups fed an atherogenic diet. With the administration of WT, BW, Glu, TC, and TG, levels decreased remarkably compared to the case group. After the 16-week experiment, serum TC and TG levels were statistically significantly lower in the WT100 and WT500 groups than in the case group (*p* < 0.05). Catechins are the primary polyphenols in tea varieties obtained from the *C. sinensis* plant using various methods, such as withering, drying, oxidation, and fermentation [23]. WT has been found to contain a higher concentration of catechins, which are natural antioxidants, compared to other types of tea such as GT and BT. Catechins are known for their potential health benefits, including their antioxidant and anti-inflammatory properties [24]. 

In our study, the weights of the animals in the WT100 and WT500 groups at the 16th week after four weeks of gavage were considerably lower than in the case group (*p* < 0.05). The tea plant has antioxidant and anti-inflammatory activity thanks to the catechins it contains [25,26]. The literature has reported that tea and its ingredients modulate glycolipids digestion, absorption, and metabolism in vitro, which may benefit obesity. Additionally, white tea, green tea, and black tea inhibited lipase activity in adipocyte cell culture and reduced lipid accumulation dose independently [27]. Another study stated that catechins increased the expression and activity of lipolytic enzymes in adipocytes and increased the release of adipose-derived glycerol [28]. Furthermore, it enhances the activity of β-oxidation-related enzymes in the liver and potentiates fatty acid transport enzymes in skeletal muscles, combined with exercise to increase β-oxidation activity [29]. Luo et al. reported that EGCG and ECG obtained from white tea extracts reduced very low-density lipoprotein (VLDL) synthesis and increased LDL receptor (LDLR) synthesis by contributing to regulating genes related to cholesterol metabolism [30]. In parallel with previous studies, our study also determined that white tea has an anti-obesity effect. Although studies in the literature show that the catechins in green tea reduce serum glucose levels, and EGCG it contains can improve insulin resistance by mimicking the effects of insulin, we did not observe any results regarding the effects of white tea on glucose in our study. 

The inflammatory response, atherosclerotic plaque record, is a distinct pathological change and is characterized by increased levels of inflammatory markers such as TNF-α, IL-6, C reactive protein (CRP), monocyte chemoattractant protein-1, as well as adhesion molecules [20,31,32]. ADAM10 and ADAM17, transmembrane proteases that are members of the ADAM family, are at the center of inflammation in the living system [33]. In the atherogenic diet-induced AS model in ApoE^−/−^ mice, atherogenic cytokine levels and ADAM10/17 activities increased while antiatherogenic cytokine levels decreased. After WT application, ADAM10/17 activities and atherogenic cytokine levels decreased, while antiatherogenic cytokine levels increased. The literature has reported that green tea consumption significantly reduces the serum levels of different inflammatory markers, such as CRP [34]. Additionally, catechins have been shown to reduce IL-6, IL-1β, and TNF-α levels while increasing the assembly of anti-inflammatory cytokines (e.g., IL-10) [35,36,37,38,39,40]. After WT administration, IL-1β, TNF-α, IL-6, and IL-12 levels decreased significantly compared to the case group, while IL-10 levels increased (*p* < 0.001). This study evaluated in vivo for the first time the effects of white tea on ADAM10 and ADAM17 enzyme activities, which play an important role in the pathophysiology of atherosclerosis. WT is thought to show anti-inflammatory activity by significantly reducing the activity of both enzymes. Ramadan et al. reported that green and black tea extracts (0.5 and 1.0 g/kg) reduced inflammation in rheumatoid arthritis-induced rats [41]. Additionally, Liu et al. stated that tea polyphenols (300 mg/kg) showed protective activity against heavy exercise-induced inflammation and tissue damage in mice, reducing TNF-α, IL-1β, and IL-6 levels [42]. In addition, it has been reported that green and black tea and their essential compounds have protective effects against inflammation and oxidative stress in a murine sepsis model [43]. The effectiveness of WT on inflammation parameters is parallel to the literature. In our study, it is thought that WT may have demonstrated its anti-inflammatory activity by reducing ADAM10/17 activities. Foam cell formation in atherosclerotic plaque formation begins with the migration and oxidation of LDL-C to the tunica intima region. After WT administration, oxLDL, LOX-1, Lp-PLA2, and MPO levels were significantly lower than the case group, while PON-1 levels were higher (*p* < 0.001). Wahyudi et al. reported that in human umbilical vascular endothelial cell culture, the flavonoids contained in green tea suppressed TNF-α production by inhibiting IKB kinase and reducing NF-κB activity and oxLDL levels [44]. Inami et al. showed that a healthy population given tea catechin supplements had lower oxLDL levels after four weeks than those who did not receive catechin supplements [45]. Shi et al. found that tea saponins reduced MPO levels in mice with alcohol-induced gastric mucosal damage [46]. Albuquerque et al. showed that MPO levels decreased with green tea treatment in neutrophils of rats fed a cafeteria diet to induce obesity [47]. Balsan et al. reported that green tea consumption increased PON-1 levels in obese humans [48]. This study has revealed for the first time the positive impact of WT on ox-LDL, PON-1, and MPO, which is consistent with the results of a green tea study. Basu et al. stated that consuming green tea beverages (four cups/day) or extract supplements (two capsules/day) for eight weeks reduced oxidative stress in individuals with metabolic syndrome [49]. Furthermore, our study has also demonstrated the effects of WT on LOX-1 and Lp-PLA2 levels for the first time. 

Histologically, H&E staining revealed extensive atherosclerotic plaque in the aortic root of ApoE^−/−^ mice fed an atherogenic diet. After WT administration, the atherosclerotic plaque burden was not reduced. Kavantzas et al. reported that green tea supplementation reduced atherosclerotic plaque burden by 30% in rabbits [50]. Cai et al. reported that EGCG reduced atherosclerotic plaque burden in the atherosclerosis model induced by Porphyromonas gingivalis in ApoE^−/−^ mice [51]. Faustin et al. reported that white tea reduced the number of plaque foam cells in the abdominal aorta in a model of atherosclerosis induced by a high-fat diet in Wistar rats [52]. Minatti et al. demonstrated that green tea reduced atherosclerotic lesions in a high-cholesterol diet-induced atherosclerosis model in LDLR^−/−^ mice [53]. However, our study does not support these findings; this may be because atheroma plaque remains in the intima even if the lipid profile improves [54,55].

In our study, white tea decreased important atherogenic parameters such as total cholesterol, OxLDL, LOX-1, IL-6, and IL-12 induced by an atherogenic diet in ApoE^−/−^ mice. However, no effects on atheroma plaque formation, which is irreversible, were observed in this study. Nevertheless, this study has some limitations. White tea extract could have started to be administered to the animals before the 12th week. However, considering the large number of groups and the ARRIVE guideline, this option should have been considered. In addition to atheroma plaque morphology typing, lipid accumulation could have been investigated by oil red O staining for en face analysis. In addition to all these limitations, this study demonstrated that white tea may have essential effects against oxidative stress, dyslipidemia, and inflammation in atherosclerosis.

## 4. Materials and Methods

### 4.1. HPLC-Mediated Content Analysis and Polyphenol/Flavonoid Capacity of White Tea

One gram of WT was weighed and then added to 100 mL of double distilled water in glass bottles. In this study, tea samples were brewed at 95 °C for 12 min, filtered through a 0.45 µm filter, and prepared [18]. The phenolic composition of white tea was examined utilizing high-performance liquid chromatography with photodiode array detection (HPLC-PDA; ISO 9002 standard). A gradient program was employed, utilizing a mobile phase consisting of 70–30% acetonitrile–ultrapure water and 2% acetic acid–ultrapure water. The chromatographic analysis was performed at a flow rate of 1.0 mL/min at an oven temperature of 30 °C. Each sample was analyzed in triplicate. The findings were expressed in micrograms per gram of material (μg/g) [56]. The analysis utilized HPLC grade GA, C, EC, EGC, ECG, EGCG, and caffeine standards obtained from Sigma-Aldrich (Munich, Germany). The total polyphenol content was determined using the Folin–Ciocalteu method, which reduces phosphotungstic acid (H_3_PW_12_O_40_) to phosphotungstic blue in the primary solution. GA was used as the standard [57]. The total flavonoid content was measured by forming acid-stable complexes with the C4 keto group and C3 or C5 hydroxyl groups of flavones and flavonols by aluminum chloride (AlCl_3_). Quercetin (Q) was used as the standard [58].

### 4.2. Chemicals, Diets, and WT Preparation

The following materials were used in this study: Triton X-100, phosphate-buffered saline (PBS), and bicinchoninic acid (BCA), all acquired from Sigma–Aldrich (St. Louis, MO, USA). Atherogenic diet (D12108C) and control diet (D12104C) were purchased from Research Diet, New Brunswick, NJ, USA. The white tea (Çaykur A.Ş. Rize, Turkey) used in this study was infused at 95 °C for 12 min, passing through a 0.45 µm filter, and freshly prepared daily.

### 4.3. Animals 

The Ethical Committee of the Faculty of Medicine at Karadeniz Technical University approved the experimental protocols (approval number: 2022-54, date: 8 December 2022). The experiments complied with the guidelines outlined for the care and use of laboratory animals, as specified by the National Institutes of Health. Additionally, the experiment adhered to the ARRIVE guidelines for reporting experiments involving animals [59,60]. These guidelines were followed meticulously to ensure the ethical and responsible treatment of the animals involved in the experiment. Mice deficient in the ApoE gene, one of the LDLR recognition proteins, have high serum cholesterol levels. Just like in human atheroma plaques, the whole spectrum of lesions was observed in ApoE-deficient mice. Therefore, it is widely used in experimental atherosclerosis studies [61]. We purchased C57BL/6J mice and ApoE^−/−^ mice from the Surgical Research Center at Kardeniz Technical University in Trabzon, Turkey. The mice were obtained at the age of 6–8 weeks. The mice were housed in a controlled environment with a 22–23 °C temperature and a 12 h light/dark cycle. 

### 4.4. Control of Food Intake

Four animals were housed in each cage. Fresh diet (≈6 g per mouse per day) was placed in the cages. Before the fresh feed was placed in the cage, the remaining feed was weighed and removed from the cage. Accordingly, the amount of feed consumed by all cages for 16 weeks was recorded.

### 4.5. Experimental Groups and Procedures

There were five experimental groups, each with eight randomly selected mice. The control group consisted of C57LB/6J mice (n = 8), while the sham, case, WT100, and WT500 groups included Apo E^−/−^ mice (n = 32) (Table 6). 

The animals were fed at a rate of 3–6 g/mouse per day throughout the dietary protocol. During the study, the mice were weighed every four weeks, and their weights were recorded. At the end of the diet protocol, the animals were humanely euthanized using 80 mg/kg ketamine–10 mg/kg xylazine, followed by decapitation and trunk blood sampling. Body blood was collected in a tube, and sera were collected and stored at −80 °C until the analysis day. In our study, aortic arch and aortic root dissection were performed, which are the regions where atherosclerotic plaque burden was observed the most (Appendix A). The aortic root was removed and fixed for histopathologic examinations. Aortic arch tissue was saved at −80 °C for biochemical investigations.

### 4.6. Measurement of Serum Routine Biochemistry Parameters

The serum samples from the mice were analyzed for levels of Glu, TG, TC, ALT, AST, and BUN using an enzymatic colorimetric method on the Beckman Coulter Analyzer AU 5800 (Brea, CA, USA) at the Clinical Medical Biochemistry Laboratory in Karadeniz Technical University.

### 4.7. Determination of Serum Oxidative Stress and Inflammation Markers 

The levels of serum IL-1β (BLS-1271Mo, BostonChem, Boston, MA, USA), PLA2 (E0267Mo, BT-LAB, Shanghai, China), ox-LDL (EM0400, FineTest, Wuhan, China), LOX-1 (EM50RB, Thermo-Fisher, Waltham, MA, USA), and PON-1 (E0509Mo, BT-LAB, Shanghai, China) were accurately measured using a commercial enzyme-linked immunosorbent assay (ELISA) kits following the manufacturer’s protocol. For PLA-A2 and PON-1, 40 µL of serum was carefully added to each well, while for IL-1β, oxLDL, and LOX-1, 100 µL of serum was precisely added. An equal standard amount was added to each well, and the sandwich ELISA protocol was meticulously applied. The colorimetric measurement was accurately performed at 450 nm using a Spectra-Max Paradigm (Molecular Devices, San Jose, CA, USA). The results are expressed per pg/mL or ng/mL.

### 4.8. Aortic Arch Homogenization 

The aortic arch tissue was prepared by immersing it in PBS with 0.01% Triton X-100 buffer. The mixture was then homogenized at 6000 rpm for 120 s on ice. Subsequently, 20 s of sonication at 130 Watts and 20 kHz was performed using a Sonics–Vibracell sonicator (Newtown, CT, USA). The homogenates were then centrifuged at 15,000× *g* for 15 min using an AllegraTM 64R centrifuge by Beckman Coulter (Brea, CA, USA), and the resulting supernatant was collected. Finally, a protein investigation was carried out using a commercial BCA kit. 

### 4.9. Investigation of Atherogenic and Antiatherogenic Parameters at the Aortic Arch

The levels of TNF-α (BLS-1395Mo, BostonChem, Boston, MA, USA), IFN-γ (BLS-1132Mo, BostonChem, Boston, MA, USA), IL-6 (BLS-1157Mo, BostonChem, Boston, MA, USA), IL-12 (BLS-9395Mo, BostonChem, Boston, MA, USA), MPO (BLS-1662Mo, BostonChem, Boston, MA, USA), IL-10 (BLS-1143Mo, BostonChem, Boston, MA, USA), and Ox-LDL (EM0400, FineTest, Wuhan, China) were measured using a sandwich ELISA using specific commercial kits. To perform the assay, 100 µL of supernatant and standard solutions were added to the wells of the appropriate antibody-coated ELISA plates. The plates were then incubated and washed according to the manufacturer’s instructions. Following the incubation and washing steps, a colorimetric measurement was performed at 450 nm using a Spectra-Max Paradigm microplate reader from Molecular Devices (San Jose, CA, USA). The absorbance readings obtained at 450 nm were used to determine the concentrations of the respective analytes in the samples according to the standard curves generated using the provided standards. The results are expressed per mg of protein.

### 4.10. Determination of ADAM10 and ADAM17 Activity at the Aortic Arch

Based on fluorometric measurements, we used commercial kits to measure the activities of ADAM10 (AS72226, ANASPEC, Fremont, CA, USA) and ADAM17 (AS72085, ANASPEC, Fremont, CA, USA) in the aortic arch. We used a FRET peptide substrate labeled with a fluorophore and quencher (QXL™ 520) for the kinetic measurements with excitation/emission of 490 nm/520 nm. These measurements were performed using the Spectra-Max Paradigm (Molecular Devices, San Jose, CA, USA).

### 4.11. Histological Analysis in Aortic Root 

The aortic root tissues from all groups were fixed in a 10% formaldehyde solution, dehydrated, cleared with xylene, and embedded in paraffin blocks. Subsequently, 4–5 μm thickness sections were obtained from the paraffin blocks and subjected to routine hematoxylin and eosin (H&E) staining.

The atherosclerotic lesions were classified as follows: no lesion; type I, early fatty streak: up to ten (10) foam cells are observed in the intima in each section; type II, regular fatty streak: more than ten foam cells are observed in the intima in each section; type III, mild plaque: foam cells extend medially, and/or the lesion is covered by a fibrotic cap; type IV, moderate plaque: lesion infiltrates the media, progressive lesion with fibrosis without structural loss; type V, severe plaque: T. media is severely damaged. The elastic laminae are broken, cholesterol crystals are visible, and mineralization and/or necrosis are observed [66,67]. It was also calculated by measuring the lesion area relative to the entire cross-sectional area [68].

### 4.12. Statistical Analysis

Before starting scientific research, it is essential to determine the minimum number of units (subjects, patients, experimental animals, etc.) required to obtain clinically and statistically significant results during the planning phase of the study (theoretical power analysis) and to demonstrate the achieved power of a completed study (experimental power analysis). An independent sample multivariate one-way analysis of variance (MANOVA) analysis is used to detect a significant effect among groups based on mean and standard deviation of TG (mg/dL) serum biochemical results. With a minimum of 80% statistical power and a significance level of 0.05, the minimum sample size required should be 30, with each group having a minimum of 6 mice [69]. During the data pre-processing stage, it was observed that the continuous variables in the dataset followed a multivariate normal distribution based on the Henze–Zirkler and Doornik–Hansen tests (*p* > 0.05). Missing values in the dataset were imputed using the random forest imputation method. The dataset was examined to identify and handle outliers/extreme values. According to the test results, three outlier/extreme values were removed from the dataset. The categorical variables in the dataset are presented as counts/percentages. Since the multivariate normality assumption was met, continuous variables were described as mean ± standard deviation. The Pearson chi-square test was applied for categorical variables, while ANOVA was conducted for continuous variables. A significance level of 0.05 was used for all tests. The tables presented in the findings were prepared using Microsoft Word (Microsoft, Redmond, WA, USA), and the figures were prepared using OriginPro 2024b (OriginLab, Northampton, MA, USA). 

## 5. Conclusions

This study demonstrated that white tea could effectively reduce the progression of atherosclerosis-related dyslipidemia, oxidative stress, and inflammation. Previous studies have reported that white tea has protective effects against various CVDs. In this study, the effects of white tea on ADAM10/17 enzyme activities, which play an important role in the atherosclerosis process, were investigated for the first time, and it was found that it decreased the activity of these enzymes. It is predicted that these enzymes may have anti-inflammatory activity due to decreased activity. However, it did not effectively change the shape of atherosclerotic plaques. Mice treated with two doses of 100 mg/kg and 500 mg/kg showed weight loss, lower lipid levels, and reduced oxidative stress and inflammation. As expected, higher doses of white tea were more effective in some parameters than lower doses. In this model, one of the most appropriate experimental animal models for atherosclerosis pathology in humans was used, with hyperlipidemia-induced atheroma plaque development and increased oxidative stress and inflammation during this development. Therefore, it is thought that including white tea in the daily routine may help prevent atherosclerosis-related oxidative stress and inflammation.

## Figures and Tables

**Figure 1 pharmaceuticals-17-01699-f001:**
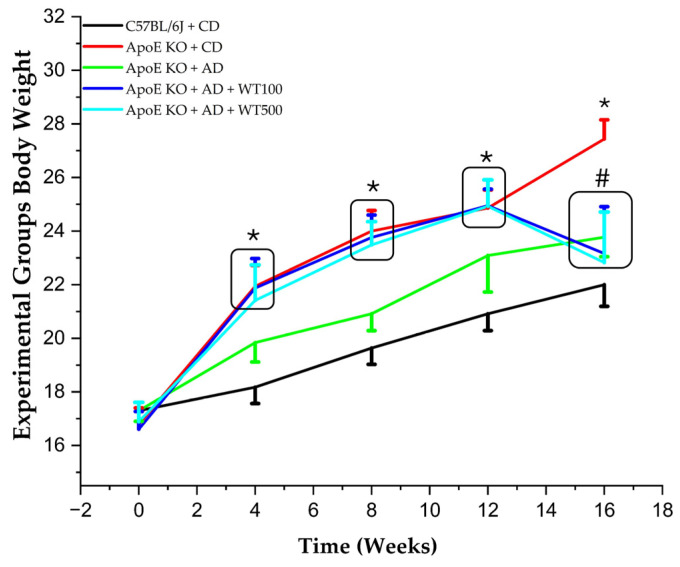
Experimental group body weight (BW) changes. **Oral gavage start:** 12th week. *, statistically significant compared to control (*p* < 0.05); #, statistically significant compared to both case and control group (*p* < 0.05) (n = 8). After determining differences between means with MANOVA, ANOVA post hoc range tests, and pairwise multiple comparisons, we determined which means differed. **CD:** control diet, **AD:** atherogenic diet, **ApoE KO:** apolipoprotein E knockout, **C57BL/6J:** wild type, **WT100:** 100mg/kg white tea, **WT500:** 500 mg/kg white tea.

**Figure 2 pharmaceuticals-17-01699-f002:**
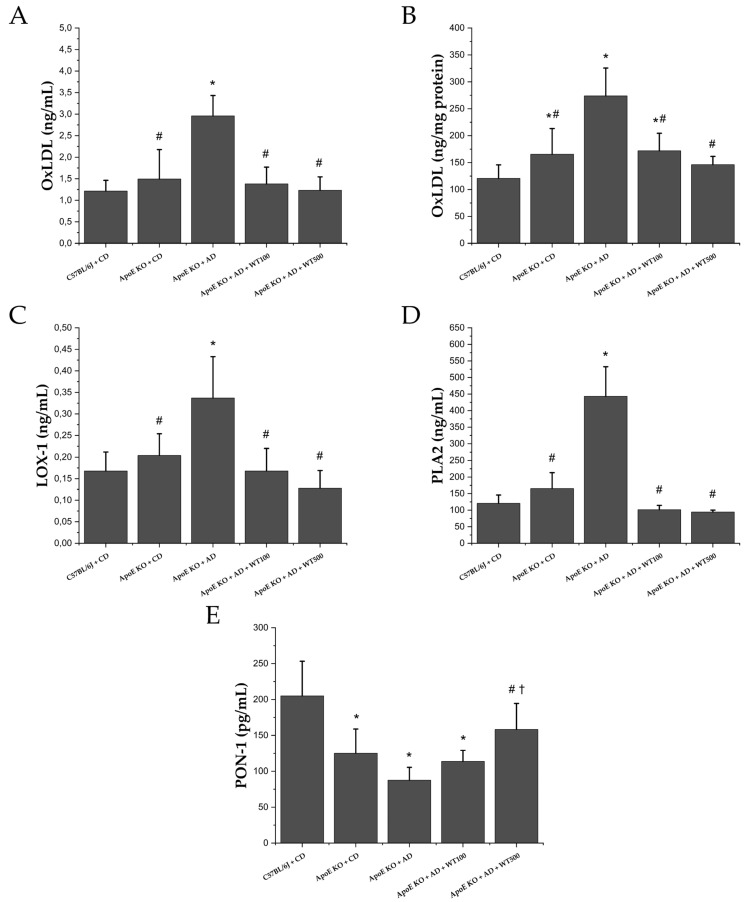
Oxidative stress markers, (**A**) oxidized-low density lipoprotein (serum), (**B**) oxidized-low density lipoprotein (aortic arch), (**C**) lectin-like oxidized LDL receptor -1(serum), (**D**) phospholipase A-2 (serum), (**E**) paraoxanase 1 (serum). *: There is a statistically significant difference compared to the control group, #: There is a statistically significant difference according to the case group, †: There is a statistically significant difference compared to the WT100 groups (*p* < 0.05) (n = 8). Multiple comparisons via multiple univariate ANOVA with Bonferroni correction after multivariate analysis of variance. **CD:** control diet, **AD:** atherogenic diet, **ApoE KO:** apolipoprotein E knockout, **C57BL/6J:** wild type, **WT100:** 100mg/kg white tea, **WT500:** 500 mg/kg white tea.

**Figure 3 pharmaceuticals-17-01699-f003:**
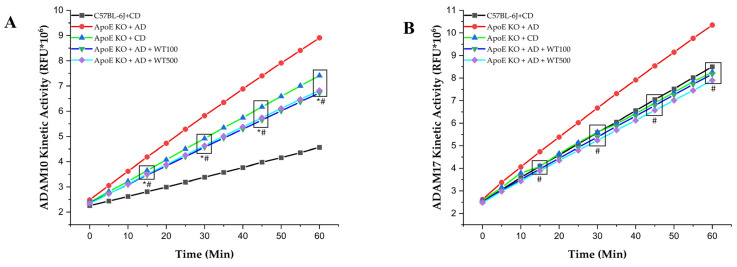
Aortic arch, (**A**) A disintegrin and metalloprotease 10(ADAM10) kinetic activity results, (**B**) A disintegrin and metalloprotease 17 (ADAM17) kinetic activity results. *, statistically significant compared to control and case (*p* < 0.05) (n = 8). #, statistically significant compared to case (*p* < 0.05) (n = 8). After determining the differences between the means with MANOVA, ANOVA post hoc range tests, and multiple pairwise comparisons, we determined which means differed. **RFU:** relative fluorescence units, **CD:** control diet, **AD:** atherogenic diet, **ApoE KO:** apolipoprotein E knockout, **C57BL/6J:** wild type, **WT100:** 100mg/kg white tea, **WT500:** 500 mg/kg white tea.

**Figure 4 pharmaceuticals-17-01699-f004:**
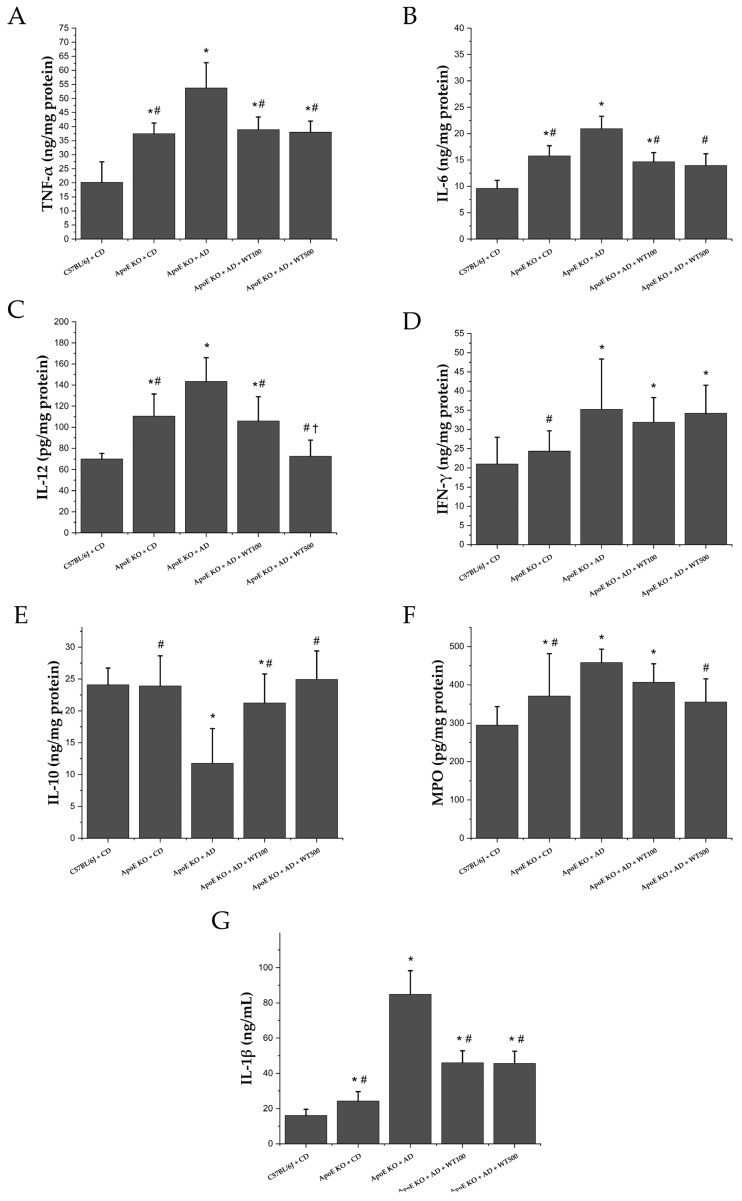
Inflammation markers: (**A**) tumor necrosis factor-α (aortic arch), (**B**) interleukin-6 (aortic arch), (**C**) interleukin-12 (aortic arch), (**D**) interferon γ (aortic arch), (**E**) interleukin-10 (aortic arch), (**F**) myeloperoxidase (aortic arch), (**G**) interleukin-1β (serum). *: There is a statistically significant difference compared to the control group, #: There is a statistically significant difference according to the case group, †: There is a statistically significant difference compared to the WT100 groups (*p* < 0.05) (n = 8). Multiple comparisons via multiple univariate ANOVA with Bonferroni correction after multivariate analysis of variance. **CD:** control diet, **AD:** atherogenic diet, **ApoE KO:** apolipoprotein E knockout, **C57BL/6J:** wild type, **WT100:** 100 mg/kg white tea, **WT500:** 500 mg/kg white tea.

**Figure 5 pharmaceuticals-17-01699-f005:**
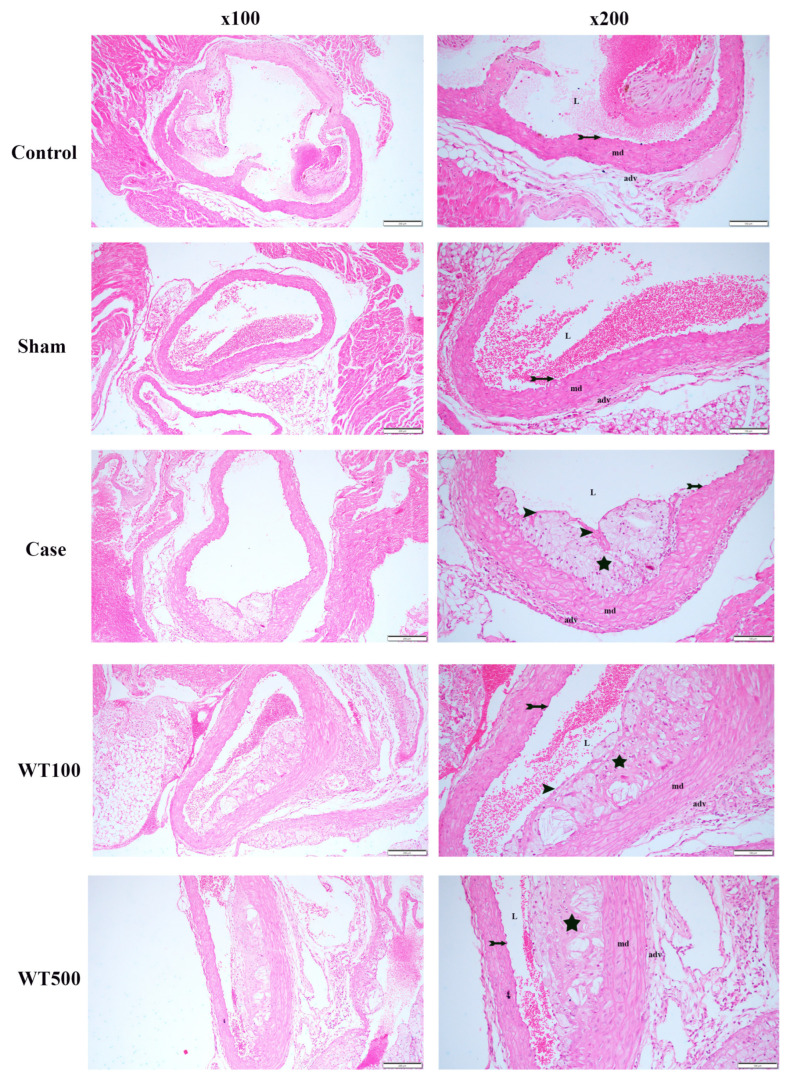
Representative light microscopic image of sections of aortic root tissue stained with H&E, n = 5, (100x, scale bar: 200 µm; 200x, scale bar: 100 µm). L: Lumen, md: Tunica media, adv: Tunica adventitia, tailed arrow: endothelium, star: atherosclerotic lesion, arrow head: fibrotic cap.

**Table 1 pharmaceuticals-17-01699-t001:** HPLC-DAD analysis of phenolic constituents of white tea.

Number	Compound Name	Retention Time (Min)	Concentration (n = 2, µg/g)
1	Gallic acid	3.202	5.55 ± 0.61
2	Epigallocatechin	4.499	261.4 ± 3.59
3	Catechin	5.488	-
4	Caffeine	7.750	101.51 ± 2.61
5	Epigallocatechin 3-gallate	8.678	119.8 ± 2.31
6	Epicatechin	9.800	-
7	Epicatechin 3-gallate	15.916	46.36 ± 1.70

**Table 2 pharmaceuticals-17-01699-t002:** Total polyphenol and flavonoid content of WT.

**White Tea** **Antioxidant** **Capacity**	**Total Polyphenol Content** **(mg GA/g dry WT)**	**Total Flavonoid Capacity** **(mg Q/g dry WT)**
	4.43 ± 0.24	157.1 ± 6.87

**Table 3 pharmaceuticals-17-01699-t003:** Food intake results.

					Food Intake g/day/mouse (X)					
Cage No.	n	Mice	Diet	Group	0–4 Weeks	4–8 Weeks	8–12 Weeks	Cage No.	Group	12–16 Weeks
**1**	4	C57BL/6J	Control	** *Control* **	5.68	5.52	5.35			5.73
**2**	4	C57BL/6J	Control		5.54	5.58	5.45			5.34
**3**	4	ApoE^−/−^	Control	** *Sham* **	5.25	5.43	5.40			5.23
**4**	4	ApoE^−/−^	Control		5.38	5.60	5.30			5.94
**5**	4	ApoE^−/−^	Atherogenic		5.70	5.55	5.15	**8**	** *Case* **	5.27
**6**	4	ApoE^−/−^	Atherogenic		5.51	5.74	5.27	**10**		5.58
**7**	4	ApoE^−/−^	Atherogenic		5.90	5.45	4.91	**6**	** *WT100* **	5.54
**8**	4	ApoE^−/−^	Atherogenic		5.40	5.27	5.53	**7**		5.28
**9**	4	ApoE^−/−^	Atherogenic		5.43	5.61	5.00	**5**	** *WT500* **	5.35
**10**	4	ApoE^−/−^	Atherogenic		5.74	5.36	5.18	**9**		5.49

**Table 4 pharmaceuticals-17-01699-t004:** Serum biochemistry results.

Parameters (X ± SD)	Control	Sham	Case	WT100	WT500
	*(C57BL/6J + CD)*	*(ApoE KO + CD)*	*(ApoE KO + AD)*	*(ApoE KO + AD + WT100)*	*(ApoE KO + AD + WT500)*
**Glu (mg/dL)**	126.8 ± 33.3	170 ± 23.4	180.8 ± 37.1	162.5 ± 33.3	172 ± 40.5
**TG (mg/dL)**	78.4 ± 16.8	69.1 ± 5.8 **^#^**	107 ± 26.5 *	52.2 ± 10.7 ***^,#^**	71.7 ± 14.5 ***^,†^**
**TC (mg/dL)**	81.2 ± 7.2	338.2 ± 25 ***^,#^**	1942 ± 195.9 *	958 ± 112.1 ***^,#^**	708.7 ± 127 ***^,#,†^**
**ALT (U/L)**	30 ± 3.1	44.2 ± 4.2	42.1 ± 11.4	33.7 ± 6.2	43.4 ± 30.2
**AST (U/L)**	148.8 ± 40.3	217.1 ± 68.7	214.6 ± 48.7	201.1 ± 43.7	221.1 ± 112.3
**BUN (mg/dL)**	22.8 ± 5.4	29.1 ± 4.8 *	28.6 ± 3 *	32 ± 3.6 *	28.8 ± 3.4 *

*****: There is a statistically significant difference compared to the control group, **#**: There is a statistically significant difference according to the case group, **†**: There is a statistically significant difference compared to the WT100 groups (*p* < 0.05) (n = 8). Multiple comparisons by multiple univariate ANOVA with Bonferroni correction after multivariate analysis of variance. **CD:** control diet, **AD:** atherogenic diet, **ApoE KO:** apolipoprotein E knockout, **C57BL/6J:** wild type, **WT100:** 100mg/kg white tea, **WT500:** 500 mg/kg white tea, **Glu:** glucose, **TG:** triglycerides, **TC:** total cholesterol, **ALT:** alanine transaminase, **AST:** aspartate transaminase, **BUN:** blood urea nitrogen.

**Table 5 pharmaceuticals-17-01699-t005:** Atherosclerotic lesion percent.

	*Lesion Area/Whole Cross-Sectional Area ×* 100 (*X ± SD*)
**Control**	0
**Sham**	0.3 ± 0.1
**Case**	27.5 ± 10.7
**WT100**	20.9 ± 15.6
**WT500**	24.2 ± 14.2

**Table 6 pharmaceuticals-17-01699-t006:** Experimental groups.

	Control	Sham	Case	WT100	WT500
**C57BL-6J Mice**	+	−	−	−	−
**ApoE^−/−^ Mice**	−	+	+	+	+
**Control Diet**	+	+	−	−	−
**Atherogenic Diet**	−	−	+	+	+
**Administration**	0.9% NaCl by o.g. for the last 28 days.	0.9% NaCl by o.g. for the last 28 days.	0.9% NaCl by o.g. for the last 28 days [62].	100 mg/kg WT by o.g. for the last 28 days [63,64].	500 mg/kg WT by o.g. for the last 28 days [65].

## Data Availability

All data generated or analyzed during this study are included in this published article.

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
