# Peer review of "White Tea Reduces Dyslipidemia, Inflammation, and Oxidative Stress in the Aortic Arch in a Model of Atherosclerosis Induced by Atherogenic Diet in ApoE Knockout Mice"

_pharmaceuticals, 2024, doi:10.3390/ph17121699_

Round 1

Reviewer 1 Report (Previous Reviewer 4)

Comments and Suggestions for Authors

The authors have resubmitted their manuscript following its initial rejection. The two primary reasons for the rejection were: 1) the lack of quantification of atherosclerosis, and 2) the potential confounding factor of weight loss, which might be attributed to a reduced intake of an atherogenic diet. This reduction could partially explain the observed decrease in dyslipidemia, and consequently of inflammation, and oxidative stress.

In their revised manuscript, the authors have added quantification of atherosclerosis in the aortic root, which I consider the minimum acceptable addition for this paper. However, the rebuttal and revised manuscript fail to provide quantification data on diet consumption across the different groups. Consequently, this major concern remains unaddressed.

Minor comments:

Additionally, there is an issue with Table 4 appearing twice: one table lists aortic root plaque quantification, and the other lists the experimental groups. Furthermore, the heading of the table with atherosclerosis quantification is incorrect.

Author Response

Author’s Response for Reviewer 1

We are immensely grateful to the Editor for taking the time to review the article and to the reviewers for their invaluable comments. We genuinely appreciate the valuable feedback provided by all the reviewers. We have carefully incorporated all the suggestions and corrections into the manuscript.

Yours Sincerely

Mehtap ATAK (PhD)

Comment 1: In their revised manuscript, the authors have added quantification of atherosclerosis in the aortic root, which I consider the minimum acceptable addition for this paper. However, the rebuttal and revised manuscript fail to provide quantification data on diet consumption across the different groups. Consequently, this major concern remains unaddressed.

Response 1: Thank you for your contribution and suggestion. Data on food intake are presented under the heading ''4.4. Control of food intake'' and the results are presented under the heading ''2.2. Food intake results''.

Comment 2: Additionally, there is an issue with Table 4 appearing twice: one table lists aortic root plaque quantification, and the other lists the experimental groups. Furthermore, the heading of the table with atherosclerosis quantification is incorrect.

Response 2: We apologize for these mistakes. Necessary corrections have been made.

Reviewer 2 Report (Previous Reviewer 2)

Comments and Suggestions for Authors

-

Author Response

Thank you for your time for our article.

Reviewer 3 Report (New Reviewer)

Comments and Suggestions for Authors

Authors have analyzed the potential of white tea on oxidative stress, dyslipidemia, and inflammation in the development of atherosclerosis. The paper is well structured. There are a few suggestions to improve the quality of the paper:

1. Introduction lacks the information about correlation between using of assay parameters with oxidative stress, dyslipidemia, and inflammation in the development of atherosclerosis. 

2. Please include original data of HPLC-DAD.

Author Response

Author’s Response for Reviewer 3

We are immensely grateful to the Editor for taking the time to review the article and to the reviewers for their invaluable comments. We genuinely appreciate the valuable feedback provided by all the reviewers. We have carefully incorporated all the suggestions and corrections into the manuscript.

Yours Sincerely

Mehtap ATAK (PhD)

Comment 1: Authors have analyzed the potential of white tea on oxidative stress, dyslipidemia, and inflammation in the development of atherosclerosis. The paper is well structured. There are a few suggestions to improve the quality of the paper: Introduction lacks the information about correlation between using of assay parameters with oxidative stress, dyslipidemia, and inflammation in the development of atherosclerosis.

Response 1: Thank you for your contribution and suggestion. Necessary correlation has been added to the Introduction section.

‘’ Ox-LDL receptor-1 (OLR-1/LOX-1), tumor necrosis factor alpha (TNF-α) and atherogenic interleukins (IL-1β, IL-6, etc.) play an important role in the pathogenesis of atherosclerosis ……..’’

Comment 2: Please include original data of HPLC-DAD.

Response 2: Thank you for your contribution and suggestion. HPLC-DAD Results are presented in supplementary material Figure 1 and unpublished materials 1 and 2.

Reviewer 4 Report (New Reviewer)

Comments and Suggestions for Authors

The group of authors tried to evaluate the potential effects of white tea (WT) on the atherosclerosis process by reducing oxidative stress, inflammation, and dyslipidemia. The study was performed on mice samples. The results can be considered significant. However, the writing style needs to be revised entirely regarding tone, integrity, clarity, and grammar. Besides, several important details have not been mentioned in the materials and methods section, which should be revised according to the following comments.

1-      Page 2 paragraph 3: The authors named the main metabolites of white tea. However, they have not clarified which component type and concentration are different from black and green tea.

2-      Please mention in section 4.1 how did you prepare the extract before HPLC analysis

3-      In section 4.1 please mention the source and purity of standards, including GA, C, EC, EGC, ECG, EGCG, and caffeine

4-      Please provide the calibration curves for the standards in section 4.1 in the response sheet and add them to the supplementary material

5-      The title of Table 1 must be more informative. Did you have biological replicates or injection replicates to report the concentration of metabolites in the 3rd column of Table 1. Please provide a standard deviation for this measurement.

6-      In section 4.1, the HPLC mobile phase, flow rate, column type, detector, and the wavelengths to detect each metabolite should be mentioned

7-      The writing style in the result section needs to be revised. For example, section 2.1 (although it is mentioned in the method) needs an introductory sentence that the reader realizes

8- A similar problem is observed in section 2.3. Please provide an introductory sentence before jumping to  “The weight of the groups…..”. Section 2.3 should be annotated as section 2.2. 

Comments on the Quality of English Language

The writing style must be revised entirely regarding tone, integrity, clarity, and grammar.

Author Response

Author’s Response for Reviewer 4

We are immensely grateful to the Editor for taking the time to review the article and to the reviewers for their invaluable comments. We genuinely appreciate the valuable feedback provided by all the reviewers. We have carefully incorporated all the suggestions and corrections into the manuscript.

Yours Sincerely

Mehtap ATAK (PhD)

Reviewer 4

Comment 1: The group of authors tried to evaluate the potential effects of white tea (WT) on the atherosclerosis process by reducing oxidative stress, inflammation, and dyslipidemia. The study was performed on mice samples. The results can be considered significant. However, the writing style needs to be revised entirely regarding tone, integrity, clarity, and grammar. Besides, several important details have not been mentioned in the materials and methods section, which should be revised according to the following comments.Page 2 paragraph 3: The authors named the main metabolites of white tea. However, they have not clarified which component type and concentration are different from black and green tea.

Response 1: Thank you for your contribution and suggestion. In the third paragraph of page 2, the difference in the content of white tea and other types of tea is stated.

‘’White tea, which is produced with a minimum of processing, is distinguished from other types of tea by its bioactive components and contains high levels of ECGC (Atak et al., 2024).’’

Comment 2: Please mention in section 4.1 how did you prepare the extract before HPLC analysis

Response 2: Thank you for your contribution and suggestion. The preparation process of extracts before HPLC analysis was added.

‘’One gram of white tea was weighed and then added to 100 mL of double distilled water in glass bottles. In this study, tea samples were brewed at 95 °C for 10 min, filtered through a 0.45 µm filter, and prepared..’’

Comment 3: In section 4.1 please mention the source and purity of standards, including GA, C, EC, EGC, ECG, EGCG, and caffeine

Response 3: Thank you for your contribution and suggestion. The source and purity of standards including GA, C, EC, EGC, ECG, EGCG and caffeine are added.

‘’Standards: GA, C, EC, EGC, ECG, EGCG, and caffeine were HPLC grade and purchased from Sigma-Aldrich (Munich, Germany).’’

Comment 4: Please provide the calibration curves for the standards in section 4.1 in the response sheet and add them to the supplementary material

Response 4: Thank you for your contribution and suggestion. Calibration curves for the standards in section 4.1 are given and are added in the supplementary material.

Comment 5: The title of Table 1 must be more informative. Did you have biological replicates or injection replicates to report the concentration of metabolites in the 3rd column of Table 1. Please provide a standard deviation for this measurement.

Response 5: Thank you for your contribution and suggestion. The title of Table 1 was corrected as ‘HPLC-DAD analysis of phenolic constituents of white tea’. Furthermore Table 1 has been revised according to your suggestions

Comment 6: In section 4.1, the HPLC mobile phase, flow rate, column type, detector, and the wavelengths to detect each metabolite should be mentioned

Response 6: Thank you for your contribution and suggestion. In Section 4.1, the HPLC mobile phase, flow rate, column type, detector, and wavelengths used to detect each metabolite were mentioned.

‘’The phenolic composition of white tea was examined utilizing high-performance liquid chromatography with photodiode array detection (HPLC-PDA; ISO 9002 standard). A gradient program was employed, utilizing a mobile phase consisting of 70–30% acetonitrile-ultrapure water and 2% acetic acid-ultrapure water. The chromatographic analysis was performed at a flow rate of 1.0 mL/min at an oven temperature of 30°C. Each sample was analyzed in triplicate. The findings were expressed in micrograms per gram of material (μg/g).’’

Comment 7: The writing style in the result section needs to be revised. For example, section 2.1 (although it is mentioned in the method) needs an introductory sentence that the reader realizes

Response 7: Thank you for your contribution and suggestion. An explanatory introductory sentence has been added to Section 2.1.

‘’The phenolic content of white tea was assessed by HPLC-DAD analysis, with the results presented in Table 1.’’

Comment 8: A similar problem is observed in section 2.3. Please provide an introductory sentence before jumping to  “The weight of the groups…..”. Section 2.3 should be annotated as section 2.2. 

Response 8: Thank you for your contribution and suggestion. An explanatory introductory sentence has been added to Section 2.3.

‘’To evaluate the effect of white tea given with the atherogenic diet on weight change, animals were weighed every four weeks throughout the experiment and the weights of the groups are shown in Figure 1.’’

Round 2

Reviewer 1 Report (Previous Reviewer 4)

Comments and Suggestions for Authors

A food intake table has been added. Although I understand that mice were housed four per cage, this addition is welcome. However, I am not sure it is necessary to show data for each cage. It is more customary (and meaningful to me) to present the data averaged in g/day/mouse for each group. Additionally, I would like to compare the average food intake between weeks 8-12 and weeks 12-16 across groups.

Author Response

Author’s Response for Reviewer 1

We are immensely grateful to the Editor for taking the time to review the article and to the reviewers for their invaluable comments. We genuinely appreciate the valuable feedback provided by all the reviewers. We have carefully incorporated all the suggestions and corrections into the manuscript.

Yours Sincerely

Mehtap ATAK (PhD)

Comment 1: A food intake table has been added. Although I understand that mice were housed four per cage, this addition is welcome. However, I am not sure it is necessary to show data for each cage. It is more customary (and meaningful to me) to present the data averaged in g/day/mouse for each group. Additionally, I would like to compare the average food intake between weeks 8-12 and weeks 12-16 across groups.

Response 1: Thank you for your contribution and suggestion. Table 3 has been rearranged according to your suggestions.

Reviewer 4 Report (New Reviewer)

Comments and Suggestions for Authors

The manuscript is improved after revision.

Author Response

Author’s Response for Reviewer 4

We are immensely grateful to the Editor for taking the time to review the article and to the reviewers for their invaluable comments. We genuinely appreciate the valuable feedback provided by all the reviewers. We have carefully incorporated all the suggestions and corrections into the manuscript.

Yours Sincerely

Mehtap ATAK (PhD)

Reviewer 4

Comment 1: The manuscript is improved after revision.

Response 1: We are grateful for your contributions to this article.

This manuscript is a resubmission of an earlier submission. The following is a list of the peer review reports and author responses from that submission.

Round 1

Reviewer 1 Report

Comments and Suggestions for Authors

The study design was good, but the introduction should be improved towards the specific diseases rather than the generalized CVD methodology, and the conclusion can be improved to improve the quality of the manuscript. The following comments need to be addressed to improve the quality of the manuscript.

1.      In line 72, the author mentioned, “Studies have shown that white tea exhibits higher anti-elastase, anti-collagenase, and antioxidant properties than green tea,” but didn’t cite any references.

2. In line 80, the author generalized the statement about CVD: “The lack of effective treatments for CVD is a significant burden on society and governments.” Is it justifiable because a large number of diseases come under CVD

3. The author should abbreviate the abbreviation while mentioning it at first, like “MANOVA.”

4. In line 328, the author mentioned, “ For the first time in the literature, the effects of any tea variety on ADAM10 and ADAM17 enzyme activities were evaluated within the scope of this study”. Can the author reassess and check the literature, for example 

1. https://doi.org/10.1002/mnfr.201300275

2. Zhao B. Green Tea Polyphenols Protect Neurons against Alzheimer’s Disease and Parkinson’s Disease. Micronutrients and Brain Health. 2009 Oct 6:255.

5. The author should mention the novelty of this study

6. The author should check the grammar throughout the manuscript

Author Response

Author’s Response for Reviewer 1 and Editor

We are immensely grateful to the Editor for taking the time to review the article and to the reviewers for their invaluable comments. We genuinely appreciate the valuable feedback provided by all the reviewers. We have carefully incorporated all the suggestions and corrections into the manuscript.

Yours Sincerely

Mehtap ATAK (PhD)

Reviewer 1

The study design was good, but the introduction should be improved towards the specific diseases rather than the generalized CVD methodology, and the conclusion can be improved to improve the quality of the manuscript. The following comments need to be addressed to improve the quality of the manuscript.

Comments 1: In line 72, the author mentioned, “Studies have shown that white tea exhibits higher anti-elastase, anti-collagenase, and antioxidant properties than green tea,” but didn’t cite any references.

Response 1: Thank you for pointing this out. Reference has been added. Please refer to line 71, reference 18.

Comments 2: In line 80, the author generalized the statement about CVD: “The lack of effective treatments for CVD is a significant burden on society and governments.” Is it justifiable because a large number of diseases come under CVD

Response 2: Thank you for pointing this out. The section between lines 41-58 has been modified as recommended.

Comments 3: The author should abbreviate the abbreviation while mentioning it at first, like “MANOVA.”

Response 3: Thank you for pointing this out. Necessary corrections were made. Please refer to line 418.

Comments 4: In line 328, the author mentioned, “ For the first time in the literature, the effects of any tea variety on ADAM10 and ADAM17 enzyme activities were evaluated within the scope of this study”. Can the author reassess and check the literature, for example – (https://doi.org/10.1002/mnfr.201300275 ,  Zhao B. Green Tea Polyphenols Protect Neurons against Alzheimer’s Disease and Parkinson’s Disease. Micronutrients and Brain Health. 2009 Oct 6:255.)

Response 4: Thank you for pointing this out. The error in the text has been corrected. Please refer to line 245.

Comments 5: The author should mention the novelty of this study

Response 5: Thank you for pointing this out. The ‘’Conclusion section’’ has been edited according to your suggestions. Please refer to lines 435-449.

Comments 6: The author should check the grammar throughout the manuscript

Response 6: Thank you for pointing this out. The article has been revised, taking into account your suggestions for grammar.

Reviewer 2 Report

Comments and Suggestions for Authors

1. For naming of subtopics, you may write as overclaimed words, eg., section 3.6. White tea reduces inflammation. The methods just check levels of inflammatory markers, not all evidence in scenario of cellular or morphological inflammation. Author may state that it reduced inflammatory marker. Please check other name of subtopics again to avoid overclaimimg.

2. In duscussion, authors may made concern in effects of epigallocatechin as the major component in the extract on atherosclerotic model via many focused mechanisms. Author referred effects of catechin that may differ to epigallocatechn by strcture or properties. Author may also compare between epigallocatechin and catechin as obviously found in other tea extracts in case of bioactive properties and effects.

3. Please clarify that why author used ApoE-/- mice as a model. 

4. Please clarufy magnification of figure 5 that marked by scale bar.

Author Response

Author’s Response for Reviewer 2 and Editor

We are immensely grateful to the Editor for taking the time to review the article and to the reviewers for their invaluable comments. We genuinely appreciate the valuable feedback provided by all the reviewers. We have carefully incorporated all the suggestions and corrections into the manuscript.

Yours Sincerely

Mehtap ATAK (PhD)

Reviewer 2

Comment 1: For naming of subtopics, you may write as overclaimed words, eg., section 3.6. White tea reduces inflammation. The methods just check levels of inflammatory markers, not all evidence in scenario of cellular or morphological inflammation. Author may state that it reduced inflammatory marker. Please check other name of subtopics again to avoid overclaimimg.

Response 1: Thank you for pointing this out. We agree with this comment and have changed the 2.5 and 2.6 subheadings.

Comment 2: In duscussion, authors may made concern in effects of epigallocatechin as the major component in the extract on atherosclerotic model via many focused mechanisms. Author referred effects of catechin that may differ to epigallocatechn by strcture or properties. Author may also compare between epigallocatechin and catechin as obviously found in other tea extracts in case of bioactive properties and effects.

Response 2: Thank you for pointing this out. We agree with this comment and have revised the 3. Discussion section accordingly. Please refer to between the lines 206-211, reference 23,24.

Comment 3: Please clarify that why author used ApoE-/- mice as a model. 

Response 3: Thank you for pointing this out. We agree with this comment and have revised the 4.3. subheading accordingly. Please refer to between lines 327-330.

Comment 4: Please clarufy magnification of figure 5 that marked by scale bar.

Response 4: Thank you for pointing this out. We agree with this comment and have revised the Figure 5 legend accordingly.

Reviewer 3 Report

Comments and Suggestions for Authors

Paper titled (White Tea Reduces Dyslipidemia, Inflammation and Oxidative Stress in the Aortic Arch in a Model of Atherosclerosis Induced by a High Cholesterol Diet in ApoE Knockout Mice) by Merve Huner Yigit et al. studied the antiinflammatory and antioxidant effect of White tea in aortic arc in a mouse model of athersclerosis induced by high chlesterol diet. This is a useful and novel study and I have these recommendations for improving the final shape.

1- Title: is long but informative and well forumated

2- Abstract need to be amended by some numerical values from the results such as % changes in values or fold changes...etc

3- key words: authors should add (mice & aortic arc)

4- Introduction: Line 37-61: this paragraph is long and needs to be shortened to focus on the topic and novelty

5- please add a paragraph on the current drugs for atherosclerosis and their drawbacks or benefits and why we need to discover new medication

6- The aim of the study is not clear please write a clear aim and how you planned to achieve it

7- Methods: write details of ethical apaporval number and date at the begin of the paragraph

8- Write the experiemental design in a separate paragraph with a separate subheading & describe the study groups appropriiately

9- describe the housing conditions in details and how did you minimized animal suffering

10- How authors collected blood is not clear

11- Methods in general lacks referneces at many occasion, please revise this.

12- Section 2.4. Measurement of serum routine biochemistry parameters : please write the code numbers of kits

also section 2.7. Determination of ADAM10 and ADAM17 activity at aortic arch

13-  Section (2.5. Determination of serum oxidative stress and inflammation markers) please describe the type of ELISA kits (direct, indirect, Sandwitch ...etc) and write code numbers for all kits

14- In statistical analysis : authors justified the use of different tests but what was the post hoc test after MANOVA?

15- In methods: authors wrote n=6 but in Table 1 (n=8) and figure, please revise

16- Please mention program used for drawing Figure 3 line graph

17- Can authors make some quanifications in figure 5? such as measuring thickness or so on

18- mention  How many images taken from each animal? how many sections? what was the criteria of evaluation of the aorta? mention in methods and figure legend

18-  Authors should give the source of chemicals, kits and antibodies completely and consistently (code, company, town, state and country) & version for software

19-  In each illustration mention the type of the presented data & the statistical test applied for analysis.

Others

20- Ensure every abbreviation is explained at the first appearnace in abstract & then in the body text

21- Use appropriate abbreviations for minutes, seconds...etc

22- Every abbreviation in figures should be explained in the figure legend to be self explanatory & stands alone.
23- Authors should confirm in methods that "every possible comparison between the study groups was considered" and apply this in results

Author Response

Author’s Response for Reviewer 3 and Editor

We are immensely grateful to the Editor for taking the time to review the article and to the reviewers for their invaluable comments. We genuinely appreciate the valuable feedback provided by all the reviewers. We have carefully incorporated all the suggestions and corrections into the manuscript.

Yours Sincerely

Mehtap ATAK (PhD)

Reviewer 3

Paper titled (White Tea Reduces Dyslipidemia, Inflammation and Oxidative Stress in the Aortic Arch in a Model of Atherosclerosis Induced by a High Cholesterol Diet in ApoE Knockout Mice) by Merve Huner Yigit et al. studied the antiinflammatory and antioxidant effect of White tea in aortic arc in a mouse model of athersclerosis induced by high chlesterol diet. This is a useful and novel study and I have these recommendations for improving the final shape.

Comment 1: Title: is long but informative and well formatted

Response 1: Thank you for your comment.

Comment 2: Abstract need to be amended by some numerical values from the results such as % changes in values or fold changes...etc

Response 2: Thank you for pointing that out. The abstract section was rearranged.

Comment 3: Keywords: authors should add (mice & aortic arc)

Response 3: Thank you for pointing that out. We modified the keywords sections.

Comment 4: Introduction: Line 37-61: this paragraph is long and needs to be shortened to focus on the topic and novelty

Response 4: Thank you for pointing this out. The section between lines 37-61 has been corrected as recommended. Please refer to between lines 48-58.

Comment 5: please add a paragraph on the current drugs for atherosclerosis and their drawbacks or benefits and why we need to discover new medication

Response 5: Thank you for pointing this out. Please refer to between lines 50-55.

Comment 6: The aim of the study is not clear please write a clear aim and how you planned to achieve it

Response 6: Thank you for pointing this out. The section between lines 78-85 was modified as recommended.

Comment 7: Methods: write details of ethical apaporval number and date at the begin of the paragraph

Response 7: Agree. We have revised it. Please refer to line 322.

Comment 8: Write the experiemental design in a separate paragraph with a separate subheading & describe the study groups appropriiately

Response 8: Agree. We have revised it. Please see section 4.3. and 4.4.

Comment 9: describe the housing conditions in details and how did you minimized animal suffering

Response 9: Thank you for pointing this out. We have modified section 4.3.

Comment 10: How authors collected blood is not clear

Response 10: Thank you for pointing this out. We have added section 4.4, line 347.

Comment 11: Methods in general lacks referneces at many occasion, please revise this.

Response 11: Thank you for pointing this out. We have added section 4.4, lines 340, 341, 343. Please reference 62-65.

Comment 12: Section 2.4. Measurement of serum routine biochemistry parameters : please write the code numbers of kits also section 2.7. Determination of ADAM10 and ADAM17 activity at aortic arch

Response 12: Thank you for pointing this out. The sections 4.6 and 4.8 have been revised as recommended.

Comment 13: Section (2.5. Determination of serum oxidative stress and inflammation markers) please describe the type of ELISA kits (direct, indirect, Sandwitch ...etc) and write code numbers for all kits

Response 13: Thank you for pointing this out. The Sections ‘’4.6’’ and ‘’4.8’’ have been revised as recommended.

Comment 14: In statistical analysis : authors justified the use of different tests but what was the post hoc test after MANOVA?

Response 14: Post-hoc tests are statistical procedures that follow a significant omnibus test, such as Wilks' lambda, to determine where differences lie between groups or variables. Depending on the research question and the assumptions of the data, there are a variety of post-hoc tests for MANOVA. Univariate ANOVAs compare the means of each dependent variable separately across groups and adjust the significance level for multiple comparisons using methods such as Bonferroni or Tukey. Here, post hoc is performed with univariate ANOVA.

Comment 15: In methods: authors wrote n=6 but in Table 1 (n=8) and figure, please revise

Response 15: We apologize but could not find the part with “n=6”.

Comment 16: Please mention program used for drawing Figure 3 line graph

Response 16: Thank you for pointing this out. The programs used to analyze the data and draw the graphs presented in the study are in section ''4.11''. Please refer to lines between 431-433.

Comment 17: Can authors make some quanifications in figure 5? such as measuring thickness or so on.

Response 17: Previous studies in the literature measured thickness in human preparations. Still, since vascular lesion typing was evaluated in the animal study score in the given literature, typing was shown in our study.

Comment 18: mention  How many images taken from each animal? how many sections? what was the criteria of evaluation of the aorta? mention in methods and figure legend

Response 18: We apologize for this omission. Added to Figure 5 legend. A total of 25 aortic roots from 5 animals in each group were evaluated histopathologically. Serial sections were taken from each animal tissue and lesion typing was performed between the first and last sections.

Comment 19: Authors should give the source of chemicals, kits and antibodies completely and consistently (code, company, town, state and country) & version for software

Response 19: Agree. We have revised it.

Comment 20: In each illustration mention the type of the presented data & the statistical test applied for analysis.

Response 20: Thank you for pointing this out. We have revised it.

Comment 21:  Ensure every abbreviation is explained at the first appearnace in abstract & then in the body text

Response 21: Agree. We have revised it.

Comment 21: Use appropriate abbreviations for minutes, seconds...etc

Response 21: Agree. We have revised it.

Comment 22: Every abbreviation in figures should be explained in the figure legend to be self explanatory & stands alone.

Response 22: Agree. We have revised it.

Comment 23: Authors should confirm in methods that “every possible comparison between the study groups was considered” and apply this in results

Response 23: Agree. We have checked it.

Reviewer 4 Report

Comments and Suggestions for Authors

This study explored the potential of white tea (WT) in reducing atherosclerosis and related inflammation, aiming to improve treatments for cardiovascular disease. C57BL/6J and ApoE-/- mice fed a control diet (CD) or high cholesterol diet (HCD), received WT treatment via oral gavage at doses of 100 mg/kg/day or 500 mg/kg/day. HPLC analysis confirmed high levels of Epigallocatechin (EGC) and Epigallocatechin 3-gallate (EGCG) in WT extracts. WT administration in the last 4 weeks of the 16-week HCD diet significantly reduced weight gain, improved dyslipidemia by lowering total cholesterol (TC) and triglycerides (TG) levels and decreased oxidative stress markers (oxLDL, LOX-1, and Lp-PLA2). WT treatment also showed reduced levels of inflammatory cytokines (IL-1ß, TNF-α, IL-6, IL-12) in both serum and aortic arch and reduced activities of the inflammation-related proteases ADAM10/ADAM17 in the aortic arch of HCD-fed mice. However, white tea did not appear to reduce atherosclerotic plaque burden in the aortic root.

Major concerns and limitations:

Mice treated with WT lost close to 10% (approximately 8.4% based on graphical estimation). Did the WT-treated mice eat much less than mice in the case group ? This by itself could explain most of the results observed.

The histological atherosclerotic plaque burden reported is only qualitative. There is no quantification. Moreover, there is no en face analysis of the aortas, which is the gold standard for atherosclerotic plaque burden quantification in such model.

Minor comments:

Why was the statistical power analysis based on BUN? BUN levels were not significantly reduced in WT groups compared to case group. The number of animals per group is rather small for a study on atherosclerosis in mice. In my opinion, we rarely see significant differences in group smaller than 10 and 12 is probably the sweet spot, although 16-20 mice per group are frequently seen.

Some English/rewriting could improve the manuscript. Pay close attention to the Material and Methods section.

What was the probe used to quantify ADM10 and ADAM17 activities?

Comments on the Quality of English Language

Could be improved.

Author Response

Author’s Response for Reviewer 4 and Editor

We are immensely grateful to the Editor for taking the time to review the article and to the reviewers for their invaluable comments. We genuinely appreciate the valuable feedback provided by all the reviewers. We have carefully incorporated all the suggestions and corrections into the manuscript.

Yours Sincerely

Mehtap ATAK (PhD)

Reviewer 4

This study explored the potential of white tea (WT) in reducing atherosclerosis and related inflammation, aiming to improve treatments for cardiovascular disease. C57BL/6J and ApoE-/- mice fed a control diet (CD) or high cholesterol diet (HCD), received WT treatment via oral gavage at doses of 100 mg/kg/day or 500 mg/kg/day. HPLC analysis confirmed high levels of Epigallocatechin (EGC) and Epigallocatechin 3-gallate (EGCG) in WT extracts. WT administration in the last 4 weeks of the 16-week HCD diet significantly reduced weight gain, improved dyslipidemia by lowering total cholesterol (TC) and triglycerides (TG) levels and decreased oxidative stress markers (oxLDL, LOX-1, and Lp-PLA2). WT treatment also showed reduced levels of inflammatory cytokines (IL-1ß, TNF-α, IL-6, IL-12) in both serum and aortic arch and reduced activities of the inflammation-related proteases ADAM10/ADAM17 in the aortic arch of HCD-fed mice. However, white tea did not appear to reduce atherosclerotic plaque burden in the aortic root.

Comment 1: Mice treated with WT lost close to 10% (approximately 8.4% based on graphical estimation). Did the WT-treated mice eat much less than mice in the case group ? This by itself could explain most of the results observed.

Response 1: 3-5 g of feed per mouse per day was freshly dropped into the cages. No metabolic cage treatment was performed in this study. However, when the amount of feed placed every day was considered as a group, it was the same as the consumption before oral gavage administration. Therefore, cachexia was not emphasized at all.

Comment 2: The histological atherosclerotic plaque burden reported is only qualitative. There is no quantification. Moreover, there is no en face analysis of the aortas, which is the gold standard for atherosclerotic plaque burden quantification in such model.

Response 2: We agreed. You are right. We added the lack of en-face analysis noted in the limitations section.  

Comment 3: Why was the statistical power analysis based on BUN? BUN levels were not significantly reduced in WT groups compared to case group. The number of animals per group is rather small for a study on atherosclerosis in mice. In my opinion, we rarely see significant differences in group smaller than 10 and 12 is probably the sweet spot, although 16-20 mice per group are frequently seen.

Response 3: We apologize for this mistake. BUN was mistakenly written instead of TG. This error has been corrected. In the power analysis based on TG data, the statistical power was 0.8114, and the sample size was 39. Since we completed our study with a sample size of 40, the statistical power was found to be 0.8287. Thank you for pointing this out. While planning the study, we investigated the number of animals used in similar model studies in the literature. In studies conducted on the HCD-induced atherosclerosis model in ApoE-/- mice, groups were often composed of 8-10 animals (1-5). Therefore, we used 8 animals in each group in our study.

  1. Fang, Y., Sang, H., Yuan, N., Sun, H., Yao, S., Wang, J., & Qin, S. (2013). Ethanolic extract of propolis inhibits atherosclerosis in ApoE-knockout mice. Lipids in health and disease, 12, 123. https://doi.org/10.1186/1476-511X-12-123
  2. Yuan, G. Q., Gao, S., Geng, Y. J., Tang, Y. P., Zheng, M. J., Shelat, H. S., Collins, S., Wu, H. J., & Wu, Y. L. (2018). Tongxinluo Improves Apolipoprotein E-Deficient Mouse Heart Function. Chinese medical journal, 131(5), 544–552. https://doi.org/10.4103/0366-6999.226063
  3. Bernardes, F. P., Batista, A. T., Porto, M. L., Vasquez, E. C., Campagnaro, B. P., & Meyrelles, S. S. (2016). Protective effect of sildenafil on the genotoxicity and cytotoxicity in apolipoprotein E-deficient mice bone marrow cells. Lipids in health and disease, 15, 100. https://doi.org/10.1186/s12944-016-0268-6
  4. Segers, F. M. E., Ruder, A. V., Westra, M. M., Lammers, T., Dadfar, S. M., Roemhild, K., Lam, T. S., Kooi, M. E., Cleutjens, K. B. J. M., Verheyen, F. K., Schurink, G. W. H., Haenen, G. R., van Berkel, T. J. C., Bot, I., Halvorsen, B., Sluimer, J. C., & Biessen, E. A. L. (2023). Magnetic resonance imaging contrast-enhancement with superparamagnetic iron oxide nanoparticles amplifies macrophage foam cell apoptosis in human and murine atherosclerosis. Cardiovascular research, 118(17), 3346–3359. https://doi.org/10.1093/cvr/cvac032
  5. Xiang, L., Wang, Y., Liu, S., Ying, L., Zhang, K., Liang, N., Li, H., Luo, G., & Xiao, L. (2024). Quercetin Attenuates KLF4-Mediated Phenotypic Switch of VSMCs to Macrophage-like Cells in Atherosclerosis: A Critical Role for the JAK2/STAT3 Pathway. International journal of molecular sciences, 25(14), 7755. https://doi.org/10.3390/ijms25147755

Comment 4: Some English/rewriting could improve the manuscript. Pay close attention to the Material and Methods section.

Response 4: The article has been revised, considering your suggestions for grammar. In addition, detailed information on the kits and chemicals used in the study was presented, and the subheadings were reorganized more understandably.

Comment 5: What was the probe used to quantify ADM10 and ADAM17 activities?

Response 5: Thank you for pointing this out. Section ''4.9.'' has been revised according to your suggestions.

Round 2

Reviewer 3 Report

Comments and Suggestions for Authors

The revised version of paper titled (White Tea Reduces Dyslipidemia, Inflammation and Oxidative Stress in the Aortic Arch in a Model of Atherosclerosis Induced by a High Cholesterol Diet in ApoE Knockout Mice) was improved compared tot he original one but stilll needs some amendments:

1- Author response 18:

Response 18: We apologize for this omission. Added to Figure 5 legend. A total of 25 aortic roots from 5 animals in each group were evaluated histopathologically. Serial sections were taken from each animal tissue and lesion typing was performed between the first and last sections.

please show clearly the number of animals & number of sections from eachanimal & images analyzed

2- The style of expressing the diet (HCD) and (CD) is not convinent
Please change one of them
For example CD can be ND or any other expression

3-  The name of group 3 is not suitable (case)? please provide a better name expressing the manipulation done to this group

4- Please provide also a clean version of the manuscript after the tracked one

5- Why authors added a red line over all figures, ? this is not clear

Author Response

Author’s Response for Reviewer 3 and Editor (Round 2)

We are immensely grateful to the Editor for taking the time to review the article and to the reviewers for their invaluable comments. We genuinely appreciate the valuable feedback provided by all the reviewers. We have carefully incorporated all the suggestions and corrections into the manuscript.

Yours Sincerely

Mehtap ATAK (PhD)

Reviewer 3

The revised version of paper titled (White Tea Reduces Dyslipidemia, Inflammation and Oxidative Stress in the Aortic Arch in a Model of Atherosclerosis Induced by a High Cholesterol Diet in ApoE Knockout Mice) was improved compared tot he original one but stilll needs some amendments:

Comment 1: ‘’Response 18: We apologize for this omission. Added to Figure 5 legend. A total of 25 aortic roots from 5 animals in each group were evaluated histopathologically. Serial sections were taken from each animal tissue and lesion typing was performed between the first and last sections.’’ please show clearly the number of animals & number of sections from eachanimal & images analyzed

Response 1: Thank you for your contribution and suggestion. Figure 5 is annotated in detail in the legend. In addition, histologic images obtained from all animals are presented as raw data.

Comment 2: The style of expressing the diet (HCD) and (CD) is not convinent, Please change one of them, For example CD can be ND or any other expression.

Response 2: Thank you for your contribution and suggestion. In the text, the high cholesterol diet (HCD) has been changed to an atherogenic diet (AD).

Comment 3: The name of group 3 is not suitable (case)? please provide a better name expressing the manipulation done to this group

Response 3: Thank you for your contribution and suggestion. The group case refers to ApoE KO mice fed an atherogenic diet. A table (Table 1) was prepared to facilitate understanding of the experimental groups. In addition, the groups have been renamed in Figures 1, 2, 3, and 4 to avoid confusion.

Comment 4: Please provide also a clean version of the manuscript after the tracked one.

Response 4: Thank you for your contribution and suggestion. We have attached a file where you can follow the changes made to the manuscript. (File name: ‘’Revised Manuscript for Reviewers and Editor (pharmaceuticals-3261416) .doc ‘’).

Comment 5: Why authors added a red line over all figures, ? this is not clear.

Response 5: Thank you for your contribution and suggestion. We did not mark the red lines you see on the figures. While preparing the article in the 1st revision following the journal format, the system automatically marked it because its place was changed.

Reviewer 4 Report

Comments and Suggestions for Authors

In my opinion the lack of appropriate quantification of histological atherosclerotic plaques and the lack of en face analysis is not a simple limitation. It is a major obstacle to any conclusion regarding atherosclerosis progression and the impact of the treatment. Moreover, the answer regarding the change of body weight is rather vague.

Author Response

Author’s Response for Reviewer 4 and Editor (Round 2)

We are immensely grateful to the Editor for taking the time to review the article and to the reviewers for their invaluable comments. We genuinely appreciate the valuable feedback provided by all the reviewers. We have carefully incorporated all the suggestions and corrections into the manuscript.

Yours Sincerely

Mehtap ATAK (PhD)

Reviewer 4

Comment 1: In my opinion the lack of appropriate quantification of histological atherosclerotic plaques and the lack of en face analysis is not a simple limitation. It is a major obstacle to any conclusion regarding atherosclerosis progression and the impact of the treatment. Moreover, the answer regarding the change of body weight is rather vague.

Response 1: We sincerely thank you for your contributions. We understand your concerns. This study targeted the aortic root, where plaque formation is most common. However, our reference source for histologic analysis is invaluable. There are many publications in which atheroma plaque typing has been performed using this method [1-5].  In addition, to explain without causing confusion regarding weight loss, in the last 28 days, when we administered white tea, saline was administered to the other groups by gavage. Therefore, we cannot say that weight gain decreased due to lack of feed consumption.

References

van der Vaart JI, van Eenige R, Rensen PCN, Kooijman S. Atherosclerosis: an overview of mouse models and a detailed methodology to quantify lesions in the aortic root. Vasc Biol. 2024 Apr 4;6(1):e230017. doi: 10.1530/VB-23-0017. PMID: 38428154; PMCID: PMC11046329.

Ying Z, van Eenige R, Beerepoot R, Boon MR, Kloosterhuis NJ, van de Sluis B, Bartelt A, Rensen PCN, Kooijman S. Mirabegron-induced brown fat activation does not exacerbate atherosclerosis in mice with a functional hepatic ApoE-LDLR pathway. Pharmacol Res. 2023 Jan;187:106634. doi: 10.1016/j.phrs.2022.106634. Epub 2022 Dec 24. PMID: 36574856.

van Eenige R, Ying Z, Tramper N, Wiebing V, Siraj Z, de Boer JF, Lambooij JM, Guigas B, Qu H, Coskun T, Boon MR, Rensen PCN, Kooijman S. Combined glucose-dependent insulinotropic polypeptide receptor and glucagon-like peptide-1 receptor agonism attenuates atherosclerosis severity in APOE*3-Leiden.CETP mice. Atherosclerosis. 2023 May;372:19-31. doi: 10.1016/j.atherosclerosis.2023.03.016. Epub 2023 Mar 28. PMID: 37015151.

Berbée JF, Wong MC, Wang Y, van der Hoorn JW, Khedoe PP, van Klinken JB, Mol IM, Hiemstra PS, Tsikas D, Romijn JA, Havekes LM, Princen HM, Rensen PC. Resveratrol protects against atherosclerosis, but does not add to the antiatherogenic effect of atorvastatin, in APOE*3-Leiden.CETP mice. J Nutr Biochem. 2013 Aug;24(8):1423-30. doi: 10.1016/j.jnutbio.2012.11.009. Epub 2013 Jan 18. PMID: 23337345.

Hoving LR, Katiraei S, Heijink M, Pronk A, van der Wee-Pals L, Streefland T, Giera M, Willems van Dijk K, van Harmelen V. Dietary Mannan Oligosaccharides Modulate Gut Microbiota, Increase Fecal Bile Acid Excretion, and Decrease Plasma Cholesterol and Atherosclerosis Development. Mol Nutr Food Res. 2018 May;62(10):e1700942. doi: 10.1002/mnfr.201700942. PMID: 29665623; PMCID: PMC6001637.

Round 3

Reviewer 3 Report

Comments and Suggestions for Authors

The revised version of paper titled (White Tea Reduces Dyslipidemia, Inflammation and Oxidative Stress in the Aortic Arch in a Model of Atherosclerosis Induced by a High Cholesterol Diet in ApoE Knockout Mice by Merve Huner Yigit et al was revised in an appropriate manner.

Authors solved the raised questions and I wish they consider this in their future publications & I wish them the best of luck

Reviewer 4 Report

Comments and Suggestions for Authors

My previous comments and concerns still remain. Any decent paper focusing on atherosclerosis in mice must have en face aorta analysis and better quantification of plaque by histology.